# OpenReview forum: "FairMedQA: Benchmarking Bias in Large Language Models for Medicine and Healthcare"
_ICLR.cc/2026/Conference — ICLR 2026 Conference Withdrawn Submission_

### Official Review · Reviewer_W5JX · 2025-10-31

**Soundness:** 3
**Presentation:** 3
**Contribution:** 1
**Rating:** 4
**Confidence:** 3

**Summary:**

This paper presents FairMedQA, a new benchmark for evaluating bias in medical LLMs using adversarial questions. Generated via a multi-agent framework from USMLE vignettes, it tests 12 LLMs, revealing significant biases. The study also shows that model upgrades can simultaneously improve both accuracy and fairness.

**Strengths:**

* **Novel Benchmark:** Introduces FairMedQA, a scalable, automated benchmark for detecting medical bias using adversarial testing.
* **Multi-Agent Generation:** A novel multi-agent pipeline (Generate, Fuse, Evaluate) creates high-quality, clinically consistent adversarial variants.
* **Comprehensive Evaluation:** Provides a large-scale empirical study of bias across 12 different LLMs and 6 model versions.

**Weaknesses:**

* **Limited/unclear human oversight:** The most critical aspect of the work involves the manual evaluation of the QA pairs with humans. Among the team of four individuals, only two have medical training (with unspecified credentials). Details of this process are minimally discussed, especially with some numerical results (like indicating the cross-reviewer agreements).
* **Simplified Demographics:** Bias is tested using limited binary categories for race, sex, and socioeconomic status.
* **Generation Model Dependency:** The quality of adversarial variants depends on the foundation models (like GPT-4O) used for generation.

**Questions:**

- Did each individual review all of the 4,806 vignette pairs?

---

> ### Author Response · Authors · 2025-12-03
>
> >Reviewer W5JX-Comment-1: Limited/unclear human oversight: The most critical aspect of the work involves the manual evaluation of the QA pairs with humans. Among the team of four individuals, only two have medical training (with unspecified credentials). Details of this process are minimally discussed, especially with some numerical results (like indicating the cross-reviewer agreements). Did each individual review all of the 4,806 vignette pairs?
>
> Response: Many thanks for raising the significant comment. Our four human annotators review the generated variants, focusing on three situations, including semantic fluency, clinical consistency, and answer leakage in the generated variants. Since the scenario involves multiple-choice questions and we have ground truth answers for each clinical vignette, each annotator can complete the annotation by carefully comparing the generated variants with the original vignette and its ground truth, although two annotators have without medical backgrounds. Each annotator has reviewed all 4.806 variants independently and addressed the disagreement via engaging discussion. We have preserved our annotation sheet, which records the initial annotations from each annotator and the final annotations after discussion. We will include a more detailed description of the human annotation process in our future manuscript.

---

### Official Review · Reviewer_haDW · 2025-11-11

**Soundness:** 2
**Presentation:** 2
**Contribution:** 2
**Rating:** 2
**Confidence:** 4

**Summary:**

This paper introduces FairMedQA, a counterfactual/adversarial benchmark for measuring demographic biases of LLMs on medical multiple-choice QA. The authors construct 4,806 adversarial vignette pairs derived from 801 USMLE-style questions by neutralizing demographic cues and then inserting adversarial demographic descriptions (race, sex, socioeconomic status) via a multi-agent LLM pipeline plus human review. They evaluate 12 representative models (proprietary and open-source), measure counterfactual fairness rate (CFR) and statistical parity difference (SPD), compare FairMedQA to an existing CPV baseline, and perform cross-version analyses showing that newer model versions (e.g., GPT-5 vs GPT-4.1) often improve both accuracy and fairness. Major empirical findings: FairMedQA elicits larger bias gaps than CPV, SPD across models ranges ~0.03–0.19, CFR varies by model (0.71–0.94), and upgrades frequently lead to “win–win” gains in accuracy and fairness.

**Strengths:**

* Careful dataset construction: multi-stage pipeline (Generation/Fusion/Evaluation agents) plus manual QC yields a large set of plausible counterfactual vignette pairs while limiting clinical-relevance leakage.

* Scale and breadth: 4,806 adversarial variants across three sensitive attributes evaluated on 12 diverse models and multiple version pairs; cross-version analysis is especially informative.

**Weaknesses:**

* The narrow task format (multiple choice) limits external validity: real clinical interactions are open-ended, contextual, and multimodal; results may not generalize to diagnosis dialogue or real EHR notes. The paper should more explicitly qualify this limitation (beyond the short Limitations section) and, if feasible, include a small pilot on one open-ended or free-text clinical dataset to show whether similar bias patterns appear.

* The scope of experiments is limited, focusing primarily on multiple-choice QA and three demographic dimensions (race, sex, SES) across 12 models. While these experiments are thorough within this scope, further analysis, such as existing fairness-enhanced tuning/prompting on medical scenarios, could provide more actionable insights for future works.

**Questions:**

N/A

---

> ### Author Response · Authors · 2025-12-03
>
> >Reviewer haDW-Comment-1: The narrow task format (multiple choice) limits external validity: real clinical interactions are open-ended, contextual, and multimodal; results may not generalize to diagnosis dialogue or real EHR notes. The paper should more explicitly qualify this limitation (beyond the short Limitations section) and, if feasible, include a small pilot on one open-ended or free-text clinical dataset to show whether similar bias patterns appear.
>
> Response: Many thanks for raising this important concern. We fully agree that the multiple-choice format can limit the external validity of FairMedQA, as real patient–doctor interactions are far more complex than multi-choice QA. However, medical scenarios are highly critical and complex, involving diverse conditions, clinical specialties, and decision-making objectives across the healthcare pipeline. Each scenario warrants its own benchmark and requires independent empirical investigation. Multiple-choice medical QA provides a simpler setting that tests basic medical knowledge and reasoning capabilities. Therefore, it is difficult to assume that LLMs would perform better in more complex and realistic scenarios if they already struggle in this fundamental setting.
>
> As introduced in Section 2, open-ended benchmarks such as EquityMedQA face challenges regarding automation, as they lack ground truth and require human experts for evaluation. Although using LLMs as judges is common in other domains, their reliability in the medical domain remains unverified and requires further investigation. At this stage, developing an automatic open-ended medical QA benchmark as a pilot is therefore challenging.
>
> We will include a more detailed discussion of this limitation in our revised manuscript.
>
> >Reviewer haDW-Comment-2: The scope of experiments is limited, focusing primarily on multiple-choice QA and three demographic dimensions (race, sex, SES) across 12 models. While these experiments are thorough within this scope, further analysis, such as existing fairness-enhanced tuning/prompting on medical scenarios, could provide more actionable insights for future works.
>
> Response: Many thanks for the insightful suggestion. In this work, we focus on developing the benchmark itself and benchmarking existing LLMs to highlight our primary contribution. Bias mitigation strategies and related empirical studies will be further investigated in our future work.

---

### Official Review · Reviewer_cWHL · 2025-11-11

**Soundness:** 3
**Presentation:** 3
**Contribution:** 2
**Rating:** 4
**Confidence:** 4

**Summary:**

This paper presents FairMedQA, a benchmark aimed at evaluating fairness in medical QA systems. The authors construct counterfactual variations of USMLE questions by adding demographic background information and altering sensitive attributes to assess how model predictions change. Experiments on 12 LLMs show that the dataset can reveal larger fairness gaps than previous work.

**Strengths:**

The paper addresses an important and timely topic in medical AI fairness. The idea of enriching counterfactuals with contextual demographic information rather than simple attribute substitution is reasonable. The evaluation covers multiple model families and versions, providing some useful observations.

**Weaknesses:**

**Major Concerns:**

The overall contribution appears limited. The core idea is to take existing medical QA questions, add demographic background, and flip basic sensitive attributes to create synthetic counterfactual data. This feels like a rather straightforward application of fairness testing techniques to the medical QA domain and does not demonstrate substantial novelty.

The distinction from prior work is not clearly established. After reading the related work, it is still unclear how this benchmark meaningfully differs from existing fairness datasets, for example FMBench. The paper claims stronger bias triggering capability, but the conceptual and methodological differences are not clearly argued. As a result, the contribution feels incremental.

**Other Concerns:**

- The human evaluation process is not described in sufficient detail. Although the paper mentions four annotators and reports agreement levels, the protocol itself is unclear. It is not stated how the annotation was carried out, whether a guideline was used, how disagreements were resolved, and whether any annotation interface or tool supported the process.

- The interpretation of the results regarding performance and fairness improvement seems overstated. The paper suggests that model upgrades can deliver simultaneous gains in accuracy and fairness. This should be described more cautiously, as the current findings only show such a pattern for the partial tested models. It would be more accurate to say that this was observed, rather than implying a broader trend.

- The demographic design is too simplified. The benchmark only considers White versus Black, Male versus Female, and High versus Low socioeconomic status. Such binary groupings oversimplify real demographic diversity, especially in a medical context. More nuanced or non binary demographic attributes would improve realism and external validity.

- The composition and roles of the annotators raise questions. Two of the annotators have an AI background rather than a medical background. It is unclear how these annotators ensured that clinical correctness.

**Questions:**

See weaknesses.

---

> ### Author Response · Authors · 2025-12-03
>
> > Reviewer cWHL-Comment-1: The overall contribution appears limited. The core idea is to take existing medical QA questions, add demographic background, and flip basic sensitive attributes to create synthetic counterfactual data. ... does not demonstrate substantial novelty.
>
> Response: Many thanks for your valuable comment. Technically, our adversarial generative framework builds on mature mutation testing and integrates LLM-based agents to generate clinical vignette variants for bias testing, thereby extending the application of mutation testing. Additionally, although we have released this framework to support the community, our primary contribution actually lies in the benchmark dataset itself and the empirical study that benchmarks the bias of existing LLMs in medical QA scenarios. FairMedQA fills the gap in automating effective bias benchmarking, and our empirical study reveals a potential win–win opportunity in which technical solutions can improve both model fairness and performance.
>
> >Reviewer cWHL-Comment-2: The distinction from prior work is not clearly established...
>
> Response: Many thanks for raising this concern. Our benchmark targets text-based medical QA, whereas FMBench is a visual QA benchmark, and thus, they differ in task modality and evaluation scope. Unlike existing text-based medical QA benchmarks that only modify the sensitive attribute itself (e.g., sex or race), our generation agent can inject additional background information related to the target sensitive attribute, which is more likely to influence the reasoning process of LLMs when answering medical questions. As a result, FairMedQA demonstrates stronger bias-triggering effectiveness compared with existing benchmarks such as CPV.
>
> As a benchmark, both benchmarking capacity and automation are essential properties. FairMedQA achieves significant improvements in both aspects. We will include a more detailed analysis and discussion in our revised manuscript to further clarify our contributions. We hope this can address your concern.
>
> >Reviewer cWHL-Comment-3: The human evaluation process is not described in sufficient detail...
>
> Response: Many thanks for raising this concern. We have clear annotation guidelines. Specifically, the annotation focuses on semantic fluency, clinical consistency, and the answer leakage in the generated variants. Annotators resolve disagreements through an all-participant discussion to reach a final consensus. We have preserved our annotation sheet, which records the initial annotations from each annotator and the final annotations after discussion. We will include a more detailed description of the human annotation process in our future manuscript.
>
> >Reviewer cWHL-Comment-4: The interpretation of the results regarding performance and fairness improvement seems overstated...
>
> Response: Many thanks for your comments. In previous fairness research, improving fairness often requires compromising the original model performance. As one of our main empirical findings, we observe that performance and fairness in LLMs are not inherently a zero-sum trade-off, and that a win–win outcome, where both are improved simultaneously, is achievable. However, this does not imply that every model version upgrade will lead to such an outcome. Rather, it demonstrates that achieving both goals is possible. We will revise our description in our future manuscript to present this finding more clearly.
>
> >Reviewer cWHL-Comment-5: The demographic design is too simplified...
>
> Response: Many thanks for the important suggestion. We fully agree that incorporating more nuanced and non-binary demographic attributes would improve realism and external validity. However, binary settings remain common practice in fairness research, and they still provide meaningful insights for developing fairer and more robust LLMs for medical QA. Even under a simple binary setting, all the LLMs we studied exhibited significant bias. If models cannot handle bias in binary-group scenarios, they are unlikely to succeed in more nuanced or non-binary settings. Therefore, at this stage, we adopted a binary setup to complete the full pipeline for benchmarking LLM bias in medical QA. We will discuss the limitations of this simplification and outline future extensions of FairMedQA to more comprehensive settings in our revised manuscript.
>
> >Reviewer cWHL-Comment-6: The composition and roles of the annotators raise questions...
>
> Response: Many thanks for raising this concern. Since the scenario involves multiple-choice questions and we have ground truth answers for each clinical vignette, each annotator can complete the annotation by carefully comparing the generated variants with the original vignette and its ground truth. Additionally, each annotator independently annotates all variants and participates in a discussion to resolve any inconsistencies, ensuring the correctness of the human annotations.

---

### Note · Authors · 2025-12-03

I have read and agree with the venue's withdrawal policy on behalf of myself and my co-authors.